# Densification Mechanism for the Precursor of AFS under Different Rolling Temperatures

**DOI:** 10.3390/ma12233933

**Published:** 2019-11-27

**Authors:** Xi Sun, Peng Huang, Xiaoguang Zhang, Nanding Han, Jinqin Lei, Yongtao Yao, Guoyin Zu

**Affiliations:** 1School of Materials Science and Engineering, Northeastern University, Shenyang 110819, China; sunxi0524@163.com (X.S.); neuhuangpeng@163.com (P.H.); zhangxg976@163.com (X.Z.); HanND03031212@163.com (N.H.); 2Key Laboratory of Lightweight Structural Materials, Liaoning province, Shenyang 110819, China; 3Northeast Light Alloy Co., Ltd., Harbin 150060, China; ljqwyt@163.com; 4National Key Laboratory of Science and Technology on Advanced Composites in Special Environments, Harbin Institute of Technology, Harbin 150001, China; yaoyt@hit.edu.cn

**Keywords:** aluminum foam sandwich, pack rolling, densification mechanism, composite interface morphology

## Abstract

The effect of rolling temperature on the precursor of aluminum foam sandwich (AFS) prepared by powder metallurgy through Pack Rolling method is investigated in this work. The cross-section along rolling direction of the precursors was observed. It was found that periodic corrugated morphology with micro-cracks on the composite interface as well as cracks and micro-holes among core powder particles emerged abundantly at room temperature rolling. These defects degraded with increasing rolling temperature and completely disappeared when the rolling temperature reached 400 °C. Combining with foaming ability of these precursors, the densification mechanism of core powders was discussed. Powder particles deformed with difficulty at low rolling temperature; the gap between them cannot be effectively filled through their plastic deformation. Fracture occurred in powder core layer during co-extension with the outer panel and was partly embedded by it, resulting in corrugated composite morphology at the interface. The precursors of high density and excellent bonding interface were prepared at the rolling temperature of 400 °C. A more suitable foaming condition was determined.

## 1. Introduction

As one of the very representative porous composite materials, aluminum foam sandwich panel (AFS) has many unique physical advantages such as light weight, high damping, excellent energy absorption, muffling and electromagnetic shielding, etc. [1,2,3,4]. After special design and assembly, its comprehensive performance can be applied in the many fields [5,6,7]. Taking AFS as the lining material of high-speed train for example, the requirements of vehicle lightweight design and certain strength requirements are satisfied. Meanwhile, the vibration and noise generated when the vehicle is moving at high speed are effectively eliminated, greatly improving the comfort of the personnel inside.

However, AFS has not been widely applied commercially at present. This is mainly due to the fact that the current industrial preparation technology for high quality AFS is not mature enough [8,9,10,11,12]. At present, the main preparation technology of AFS was to adopt adhesive compounding method through Melt Foaming Process. However, there were many insufficiencies in the glue layer such as vulnerability to aging, lack of high temperature endurance and low strength, which have limited its comprehensive application.

Due to the unique technological advantages of manufacturing AFS by powder metallurgy process (PM) [13,14], metallurgical bonding between panels and foam core was more conducive to be realized. Thus, more attempts have been made to explore new processes based on PM method. The mainly compound mechanism for AFS prepared by PM was thermal diffusion [15]. Specifically, after closely fitted with each other of densified core powders and metal panel outside by high pressure through machine equipment, a certain bonding strength was gained for the interface. During the later foaming process when temperature was above the melting point of powder core layer, inter-diffusion in boundary occurred and thus, the effect of high metallurgical composite was gained.

Bonding strength between panel and powder core layer as well as densification of core powders have constituted the most critical prerequisites for manufacturing AFS. Lin et al. [15] prepared a the precursor of AFS at the pressure of 8 MPa for 8 min at a temperature of 490 °C by hot pressing. The maximum expansion rate of nearly 400% for powder core layer was acquired, characterized by high quality metallurgical bonding between the panel and the foam core layer. However, it cannot meet the large-scale production in industry for the limitation of manufacturing equipment and complex production processes. Banhart et al. [1] disclosed a manufacturing technology for large-size sandwich panels in the patent. After compounding the density core powders and the metal panel by rolling, foaming temperature of more than 650 °C was adopted and thus, the AFS with size of 1 × 2 m^2^ was successfully prepared. However, the details of its preparation techniques were not disclosed at present.

Zu et al. [16] proposed another technological route for the preparation of AFS. The operation of pack rolling method was implemented after sealing the mixed metal powders in the aluminum alloy cavity. The powder core layer was densified and closely adhered to the metal faceplate, thus creating the necessary preconditions for the preparation of AFS of high expansion ratio and excellent metallurgical bonding strength with exterior panel. However, there were also some essential parameters that need to be determined before preparation of AFS industrially. Especially for the rolling temperature of precursor, there is no relevant literature about it.

In this work, densification process of core powders at different rolling temperatures was investigated by observing the morphology of composite interface and the bonding state of powder particles after rolling. The effect of rolling temperature on the foaming ability of AFS was also described, which provided a basic theory for AFS manufactured by Pack Rolling.

## 2. Experimental Part

Figure 1 shows the flow chart of AFS prepared by powder metallurgy through pack rolling method. The matrix powders for preparing aluminum foam were composed of common commercial element (manufacturer: Xing Rong Yuan, Beijing, China). The composition of the mixed powders are shown in Table 1 [17,18]. After mixing these powders for 2 h by SYH-5 three-dimensional mixer, the uniform powders distribution was obtained. It is worth mentioning that the titanium hydride was pre-oxidized in air at 470 °C for 3 h in order to decrease the adverse effect of premature release of hydrogen [19,20]. The cavity material required for the experiment was 6063 aluminum alloy. After being annealed at 400 °C for 30 min, the flat tubular cavity with a total height of 18 mm was prepared by rolling mill from the original pipe with diameter 90 mm and wall thickness 3 mm. In order to improve the composite efficiency of the dense powders with the outer panel, the inner surface was cleaned by immersed in a sodium hydroxide aqueous solution with concentration of 40 g/L for 10 min and then was immersed in dilute hydrochloric acid (115 g/L) until the inner surface reached original color of Al.

After that, the mixed metal powders were sealed in the treated flat tubular cavity and both end of flat tubular cavity were flattened and then riveted. It is worth noting that in order to release the gas present in the powder during the rolling process, there were several gaps in the sealing position. However, it is also necessary to prevent the powder from overflowing during the rolling process so that there were enough sponges in both sealing positions.

The process of pre-rolling treatment of powders in the core layer is necessary. At this stage, the powder was flowability with abundance of air inside. With low reduction ratio (5% to 10%) per rolling pass at the rolling rate of 80–100 mm/s [21], a more uniform distribution of core powders was awarded. When extension along rolling direction occurred, the pre-rolling treatment stage was accomplished. At this point, the density of core powders had reached the limitation manufactured by the pre-rolling treatment stages, and the height of production was about 14 mm.

Subsequently, the rolling bonding stage for manufacturing AFS precursor was carried out. The factors of density of powder core layer and its bonding strength with aluminum alloy panel outside were the main concern. Therefore, it is necessary to ensure a certain total deformation and choose a reasonable rolling temperature.

In this experiment, the effects of rolling temperature at 20 °C, 250 °C and 400 °C on the densification process of powder particles and the composite state of dense powders and panel were studied. After cutting with electrical-discharge cutting machine (EDM), the cross-section of samples were observed with unaided eye. They were mechanically ground with 400–2000 grit grinding paper and then polished with an oxide suspension oxide (MgO) on the deerskin; their microscopic distribution of powder particles were observed by metallurgical microscope of Olympus. A foaming test was carried out for 14 min in a common heating furnace of constant temperature of 600 °C, and the sample size was 50 mm × 50 mm × 5.5 mm. Based on the above experiment, the influence of rolling temperature on the precursor of AFS was discussed, and the related densification mechanism of powders was analyzed.

## 3. Results and Discussion

Figure 2 shows the cross-section of precursor of AFS trimmed by EDM equipment in the rolling direction. All the preparation processes were identical except for the rolling temperature. The dense powders in core layer tightly attached with the aluminum alloy panel outside after rolling, which creates the necessary precondition for the metallurgical bonding in the later foaming process. Comparing these rolled specimens with the naked eye, there were significant differences in the position of composite interface.

As shown in Figure 2a, the morphology of composite interface was poor with numerous ripples when rolled at room temperature. The ripples were deep, and their high frequency presented obvious periodic characteristics. Figure 2b shows the sample that rolled at the temperature of 250 °C. Although corrugated morphology of bonding interface also existed, the frequency and amplitude were significantly reduced compared with sample in Figure 2a. Figure 2c exhibited the sample rolled at 400 °C. The morphology of composite interface is completely smooth with nearly no obvious corrugated morphology any more.

In order to further understand the distribution state of the corrugations after cold rolling, the bonding interface between the panel and the core layer was peeled manually, and its morphology of bonding interface was observed. As shown in Figure 3, the samples were same in deformation but different in rolling passes. It can be seen that a large number of corrugated interface lines were periodically distributed along the rolling direction. Compared with the sample rolled by two passes in Figure 3a, the corrugated interface lines of sample rolled by multi-pass in Figure 3b were obviously denser. Therefore, it can be concluded that the corrugated interface between the panel and the dense powders of AFS precursor increases with rolling pass.

Figure 4 shows the cross-sections of precursor along rolling direction at different rolling stages. The samples were rolled through different passes with total rolling reduction ratios of 28.6%, 44.4% and 54.8% at room temperature and 400 °C separately. The rolling rate of about 200 mm/s was adopted. The densification process of core powders and the evolution of composite morphology between powder core layer and panel during the rolling process were observed and analyzed.

Figure 4a shows the samples that rolled at room temperature. After rolled by first pass (total reduction rate, 28.6%), the panel was tightly fitted to the powder core layer and a periodic corrugated geometry appeared at the bonding interface simultaneously. Although a certain density of the powder core layer had been obtained, large numbers of micropores and microcracks were still visible, especially those clearly shown in Figure 5a that is the metallographic microscopic image of corresponding sample in Figure 4a. When the sample was subjected to the second rolling at room temperature, the total reduction ratio reached 44.4%. The periodicity corrugated on the bonding interface became more pronounced, accompanied by emerging abundant of gaps on it. However, the density of the internal powder core layer did not increase as the rolling reduction increased. On the contrary, the density decreased due to formation of some obvious crack gaps inside. Figure 5b shows the microstructure of the corresponding sample, the cracks existed in the bonding interface and the fracture zone in powder core layer were shown more clearly. When the sample was rolled for the third time, the total reduction ratio was 54.8%. Although there was still a corrugated bonding morphology at the joint interface, there were no obvious crevices thereon. Simultaneously, the cracks originally existing in the powder core layer were healed and thus, the density obviously increased. The corresponding microstructures are shown in Figure 5c.

Similar to the samples rolled in Figure 4a, the samples in Figure 4b underwent the same rolling schedule at 400 °C. The bonding interface between the panel and the powder core layer was tightly fitted. A flatter bonding interface was acquired with nearly no obvious gaps and corrugated feature compared with the samples in Figure 4a. Meanwhile, nearly no obvious microcracks appeared in the powder core layer during all the rolling processes, which has been shown in Figure 5d,e.

In order to further explore the bonding state of powder particles inside, the corresponding high-magnification metallographic image of the sample is shown in Figure 6. It can be seen that for the sample cold rolled by first rolling pass, there were large numbers of tiny gaps among powder particles, which resulted in a decrease in the density of powder core layer, as shown in Figure 6a. After rolled by second rolling pass at room temperature, the density decreased due to formation of large numbers of cracks, which were more likely to occur next to silicon powder particles (see Figure 6b). When the sample was rolled at room temperature for the third time, no significant cracks were present except for some of the fuzzy visible composite fine lines in the powder core layer. Figure 6c–e shows the microscopic image of the samples rolled at 400 °C. During the entire rolling process, the particles were compacted tightly with each other with no obvious gaps at their boundaries. The density of the powder core layer was much higher than that of samples rolled at room temperature under same reduction rate.

The effect of rolling temperature on the composite interface was shown clearly by comparing the microscopic interface morphology in Figure 6c,f. The panel fully connected with its powder core layer for the sample rolled at 400 °C. However, a few microcracks were still detectable on the composite interface for the sample rolled at room temperature.

It is generally accepted that the preparation process of precursor is the most critical step for the manufacture of AFS. Being the prerequisite, the quality of core layer of AFS was largely determined by this stage, in particular for the foaming ability and the defects equivalence [22]. Among all the influence factors, the rolling temperature is most prominent, which directly affects the deformation of powder core layer and its compound mechanism with panel outside. Therefore, the explicitness of the mechanism of rolling temperature to densification process is of vital importance.

In the early cold rolling process, the fluidity of powders in core layer declined sharply with the increasing of powder core layer’s density. As the outer panel begin to extend, there was nearly no flow capacity for the powders inside. For the samples rolled at room temperature afterwards, although the powders in core layer were tightly connected with each other after first rolling pass, there were still numerous crevices in the powder particles of high deformation resistance. Compared with sample rolled at 400 °C under same rolling deformation, the plastic deformation of the powder particles seems more difficult to achieve and thus, the gap among them cannot be effectively filled through their plastic deformation. Especially in the vicinity of silicon particles with hard (54 HV) and brittle characteristics, large numbers of gaps existed in the periphery, which constituted of numerous defects in powder core layer. After the second cold rolling, the panel and powder core layer extended simultaneous along the rolling direction with the increase of rolling reduction. Aluminum alloy panel with good plastic deformation ability was more prone to plastic deformation. However, for the compact powder core layer inside, the form of plastic deformation occurred with more difficulty due to numerous of defects therein. Considerable fractures occurred ceaselessly during co-extension with outer panel. Especially at the area where silicon particles were abundant, numerous crackles shuttled through among them and thus, the density of powder core layer declined extremely.

Meanwhile, after the powder core layer and aluminum alloy panel were joined together by the first rolling pass, the same elongation should also be obtained in the next rolling process for the sample as a whole. However, because of high hardness (46 HV) and poor plastic deformation ability of powder core layer, it was difficult to co-deform with panel outside in a plastic deformation manner. Fracture occurred periodically during the co-extensive process with the outer layer panel in the rolling direction when the stress accumulation reaches the limitation by periodicity. Simultaneously, the aluminum alloy panel with lower hardness (27 HV) embeds itself into the crackle of powder core layer under the rolling pressure and thus, periodic undulation morphology formed at the composite interface.

However, the cracks in the powder core layer were significantly reduced after third rolling, while the plastic deformation ability of it was still poor. Owing to the existence of gaps in powder core layer after the second rolling pass, more favorable condition was created for the powder flow during the next rolling process. The dense core powders were more likely to slide along the fracture area during the third rolling pass, which not only effectively healed the cracks but also facilitated the deformation of the powder core layer during the co-extension process with the panel, rather than merely deepening the corrugated morphology at the composite interface. The densification mechanism of powders during cold rolling process is shown in Figure 7.

With the increase of rolling temperature, the plastic deformation ability of internal metal particles was significantly improved. Figure 5d–f show the samples rolled at 400 °C. The defects of microcracks were significantly reduced. Especially in the vicinity of silicon particles, there were no obvious cracks at all. Thus, both of the density of powder core layer and the bonding strength between powder particles were improved. Meanwhile, the compound effect of the bonding interface was also significantly improved with nearly no obvious corrugated geometry, which was shown in Figure 5d–f.

Figure 8 shows the foaming ability of precursor under different rolling conditions. All the samples were foamed at 600 °C for 14 min, enhancing heat transfer by using graphite plate below. The foamability is evaluated by the height expansion rate E, which can be approximately described by the following equation:E = (H − H0)/(H0 − 2b)(1)
where H is the height of aluminum foam sandwich panel after foaming. H0 is the initial height of precursor before foaming. b is the thickness of aluminum alloy plate.

It can be seen that all the samples that were rolled at room temperature showed extremely poor foaming ability. Even for the sample with rolling reduction rates of 54.8%, the expansion rate was no more than 50%. With the increase of rolling temperature, the expansion rate increased obviously. Especially for the sample that was rolled at 400 °C with reduction rates of 54.8%, an expansion rate of more than 400% was gained.

Figure 9 shows the section diagrams of foamed samples with reduction rates of 54.8% at the rolling temperatures of 20, 250 and 400 °C. It can be seen that the sample that was rolled at 20 °C showed a few crack-like cellular structures. By contrast with the precursor of same rolling condition (Figure 4a), the conclusion was drawn that the gases that derived from the decomposition of titanium hydride preferred to escape from the gaps among powder particles and were partly retained by crackle. Thus, it created an unfavorable condition for the formation of bubbles during foaming process and the original tiny crevices inside the matrix were enlarged as shown in Figure 9a. As for the sample rolled at 250 °C shown in Figure 9b, the numbers of foam cells increased markedly. However, the morphology was irregular, pores merged in large quantities. Figure 9c shows the sample rolled at 400 °C. A more uniform cell with better roundness morphology was obtained with nearly no pore mergence.

It is generally believed that a higher density of powder core layer of excellent bonding state between powder particles is the crucial necessary condition for the foaming ability and cell morphology of the aluminum foam prepared by PM method [23,24]. Under similar foaming environment, the higher the expansion rate and the better the uniform cell distribution for AFS owned were, the higher the density and the fewer the defects of powder core layer of corresponding precursor were. During the whole foaming process characterized by continuous heating process, titanium hydride decomposed continuously with the increasing of temperature. For the powder core layer characterized by higher density of excellent bonding state between powder particles under the rolling temperature of 400 °C, large quantities of gas were effectively intercepted in situ and few defects formed afterwards, leading to a higher foaming expansion rate and uniform cell morphology.

However, as plastic deformation of powder particles became more difficult with the decrease of rolling temperature, the bond strength between the powder particles weakened. Numbers of crevices among powder particles as well as cracks in the matrix appeared abundantly during the rolling process. In the foaming process afterwards, the gas decomposed by titanium hydride was easily to drain from the crevice and partly remained in the cracks. Moreover, the interlocked powder particles of poor bonding strength were prone to rupture under internal stress caused by gas accumulation with the decomposition of titanium hydride [25]. Thus, the foaming ability decreased sharply with the gas loss, and cell uniformity declined obviously for the gas transfer caused by abundant defects inside. Especially for the samples that were rolled at room temperature, the expansion rate improved limitedly even if the rolling reduction had reached 54.8%. The cellular structure was in homogeneous with nearly no foam characteristics. Thus, in order to improve the foaming ability and cell uniformity of AFS, a more reasonable rolling temperature must be determined to improve the bonding effect of core powders inside.

## 4. Conclusions

(1)In the rolling process of precursor of AFS by PM, the rolling temperature has constituted the key parameter to decide the densification process of core powders.(2)When the rolling temperature was lower than 250 °C, the interface between the plate and the core layer along the rolling direction was prone to corrugated geometry, which resulted in the anisotropy of the aluminum alloy panel. It disappeared at the rolling temperature of 400 °C. The fluctuation of bonding interface seems to decrease with the increase in rolling temperature.(3)It was difficult to improve the densification of powder core layer when the rolling temperature was lower than 250 °C. Large number of gaps existed among powder particles. Especially in the vicinity of silicon particles, numerous crackles occurred frequently along the rolling direction in powder core layer when it co-extends with the outer panel.(4)The influence of rolling temperature on the morphology of composite interface as well as the bonding state between powder particles was mainly related to the deformability of powder core layer during rolling. After rolled by certain densification at room temperature, the core powders lost their fluidity and presented poor plasticity while the plasticity of aluminum alloy panel was excellent. Fractures occurred periodically during co-extension process with the outer layer panel in the rolling direction when the stress accumulation reached the limitation intermittently. The panel was partly embedded in the crack and formed corrugated composite morphology. The morphology of composite interface as well as the bonding state of core powder particles improved significantly with the improvement of plasticity of powder core layer at the rolling temperature of 400 °C.(5)The foaming ability improved limitedly with the increasing of rolling reduction when it was rolled at room temperature. With rolling temperature rising, the foaming ability and cell uniformity improved obviously owing to higher density of powder core layer and excellent bonding state among powder particles.

## Figures and Tables

**Figure 1 materials-12-03933-f001:**
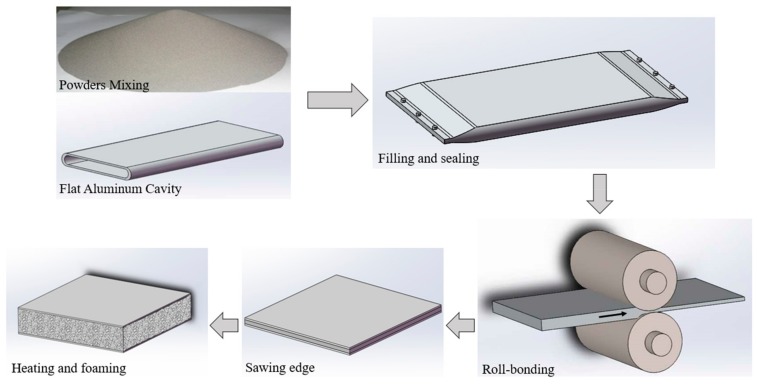
Process flow chart for preparation of AFS.

**Figure 2 materials-12-03933-f002:**
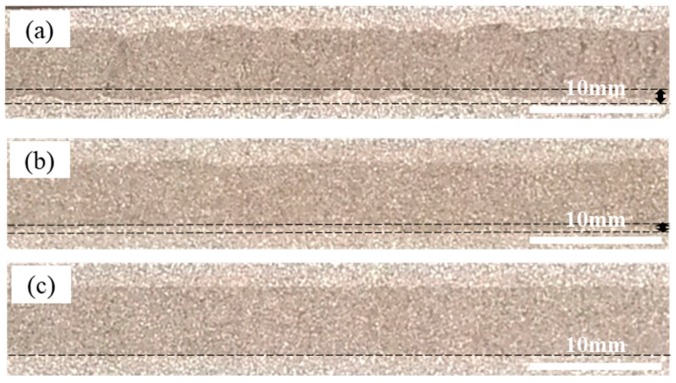
Cross-section of composite interface along rolling direction rolled at (**a**) room temperature; (**b**) 250 °C; (**c**) 400 °C.

**Figure 3 materials-12-03933-f003:**
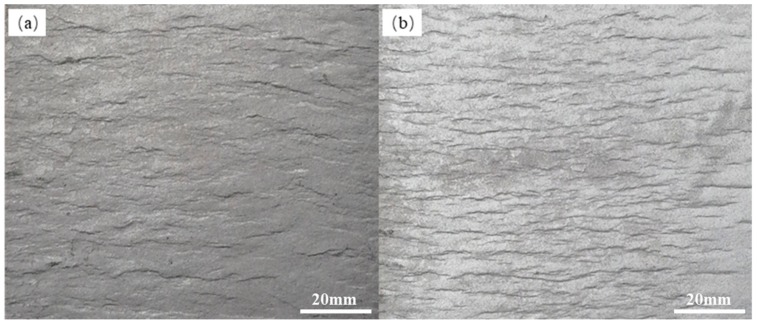
Stripped bonding interface of precursor of AFS that rolled at room temperature with rolling process of (**a**) two passes and (**b**) eight passes.

**Figure 4 materials-12-03933-f004:**
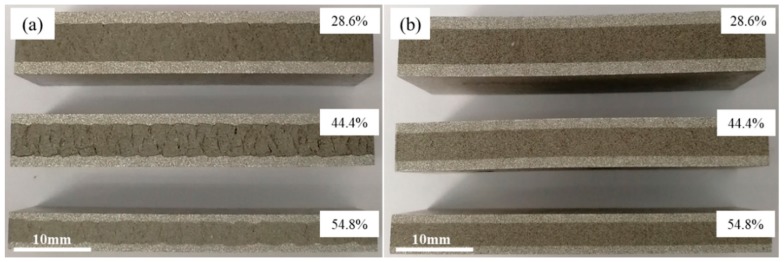
Cross-section of precursor during densification process at (**a**) room temperature; (**b**) 400 °C.

**Figure 5 materials-12-03933-f005:**
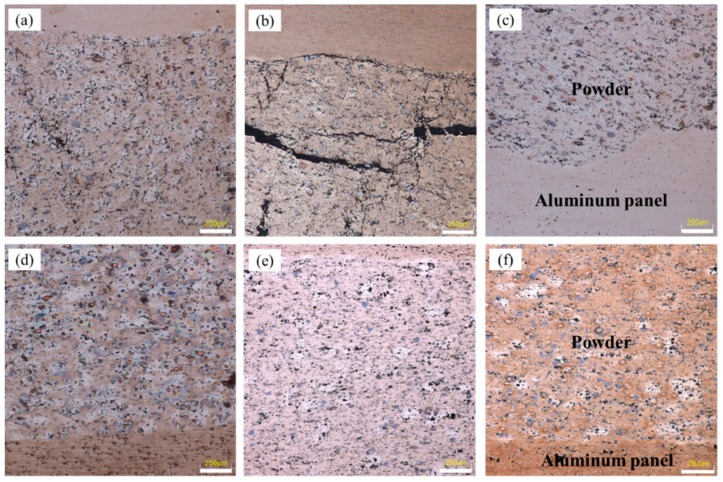
Microscopic images of dense core powders and panel at the rolling reduction ratio and temperature of (**a**) 28.6%, room temperature; (**b**) 44.4%, room temperature; (**c**) 54.8%, room temperature; (**d**) 28.6%, 400 °C; (**e**) 44.4%, 400 °C; (**f**) 54.8%, 400 °C.

**Figure 6 materials-12-03933-f006:**
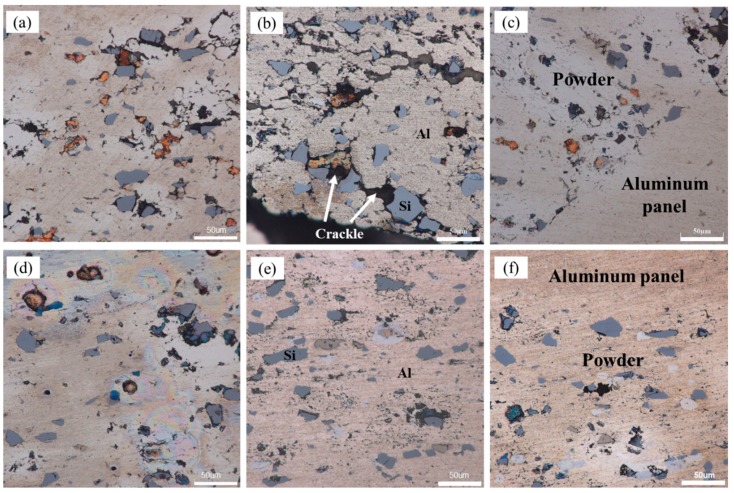
Schematic diagram of combined state among powder particles at the rolling reduction ratio and temperature of (**a**) 28.6%, room temperature; (**b**) 44.4%, room temperature; (**c**) 54.8%, room temperature; (**d**) 28.6%, 400 °C; (**e**) 44.4%, 400 °C; (**f**) 54.8%, 400 °C.

**Figure 7 materials-12-03933-f007:**
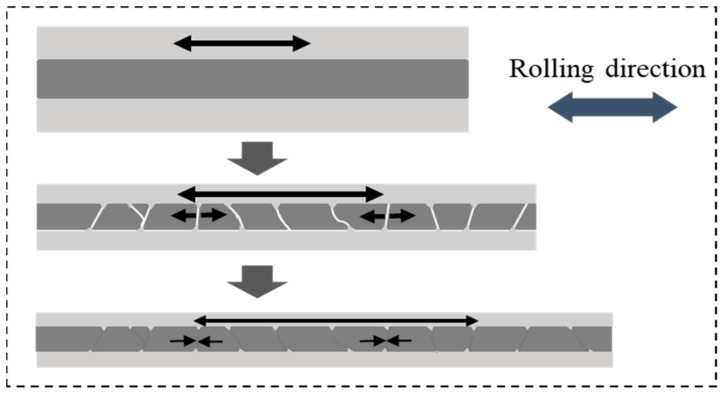
Densification mechanism of powders during cold rolling process.

**Figure 8 materials-12-03933-f008:**
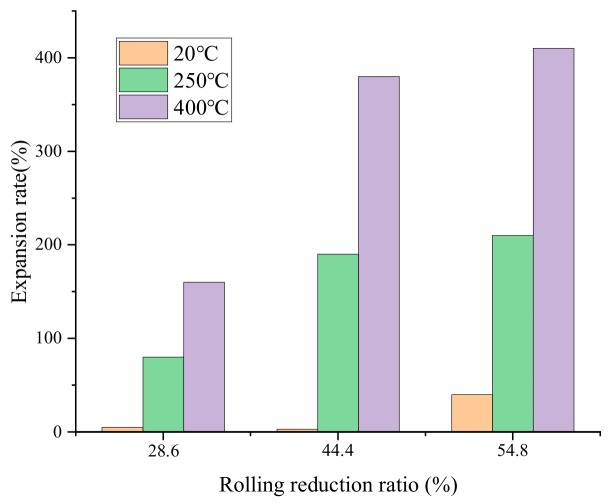
Foaming ability of precursor under different rolling conditions.

**Figure 9 materials-12-03933-f009:**
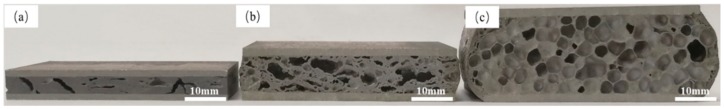
Cross-section diagram of foamed samples with reduction rates of 54.8% at the rolling temperature of (**a**) 20 °C; (**b**) 250 °C; (**c**) 400 °C.

**Table 1 materials-12-03933-t001:** Elemental composition of mixed powders.

Composition	Range Size (µm)	Purity (%)	Content
Al	<45	99.70%	85%
Si	<38	99.50%	6%
Mg	<75	99.90%	4%
Cu	<38	99.90%	4%
TiH_2_	<45	99.70%	1%

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
