# Peer review of "Densification Mechanism for the Precursor of AFS under Different Rolling Temperatures"

_materials, 2019, doi:10.3390/ma12233933_

Round 1
Reviewer 1 Report
The work carried out by Sun et al. is investigating an important topic. The story is presented in a logical and complete way. The writing is clear and systematic. The following comments need to be considered before publication in Materials
1. The state-of-the-art in the field should be described in more details to allow interested readers understanding the novelty this study possesses. In addition, more reference related to this topic should be discussed and cited.
2. The captions to the figures must give full details so that the information can be clearly understood by the reader.
3. It is advised to add mechanical properties (tensile or impact properties) to improve the quality of this manuscript.
Author Response
Dear peer reviewer.
Thank you very much for your patience, numbers of deficiencies and errors in the article have been corrected under your kindly guidance.
The state-of-the-art in the field should be described in more details to allow interested readers understanding the novelty this study possesses. In addition, more reference related to this topic should be discussed and cited.
Authors: Thank you very much for your valuable advice, I have added abundant background in the first paragraph and thus, more references were added in this article as well.
The captions to the figures must give full details so that the information can be clearly understood by the reader.
Authors: Thank you for your comment, almost all the title has been improved in the airticle. It can be more clearly understood by the reader. Grateful!
It is advised to add mechanical properties (tensile or impact properties) to improve the quality of this manuscript.
Authors: Thank you for your advice. You have made me realize my shortcomings. The mechanical properties of tensile and impact properties is very useful and I will do it latterly.
Reviewer 2 Report
Dear authors. It was nice to read your research. Please find comments and suggestions in the attached file or below.
The paper analyses the dependence of densification on rolling temperature and numbers of rollings performed on the samples (reduction rate). It presents also foaming effect in samples prepared at 3 different temperatures and 3 different reduction rates. Samples were investigated by means of optical microscopes and the naked eye. Results are interesting, scientifically logical, presented nicely in graphs, sketches, photos and with good microscopy. Paper has big potential for further investigation and can be improved by a few following suggestions:
I strongly advise to use native English speaker or full MDPI's English service or sth similar. There are smaller language errors like missing the/a, using important instead of importance, significantly instead of significant and vice versa, general instead of generally, using the wrong tense (to preparing -> to prepare…), words that can be left out (like “by people”), word however is used man times, word cross section should be everywhere cross-section etc. There are many sentences that need bigger correction so that the reader can understand what was meant (just few):
“However, there were many insufficient MISSING WORD existing for the glue layer” – is the missing word data? of 490℃, MISSING WORD? the expansion rate of nearly 400% “Being difficult of deformation for powder at low rolling temperature, the gap between powder particles was not conducive to fill.” If I understood correctly: “At low rolling temperature powder was difficult to deform, therefore the gap between powder particles was not USE ENGLISH SPEAKER” "as the plastic deformation of powder particles becoming more difficultly with rolling temperature decreasing", because it is hard to understand. If I understood it correctly: “However, as plastic deformation of powder particles became more difficult with the decrease of rolling temperature, the bond strength between the powder particles weakened.”
ABSTRACT: “when the rolling temperature was up to 400℃” – up to 400 can be many temperatures below 400 (and the test was not performed above 400, so it is better without “up to”)
INTRODUCTION (English!):
Mpa is MPa no specific details of preparation technology have been introduced about it due to technical confidentiality or some other reasons (I would leave the underlined part out since it is not a fact, the rest should be modified by native speaker) which consist of the necessary precondition for high porosity of AFS that characteristics by high metallurgical bonding strength. However, many detailed technical parameters still need to be determined for the preparation of AFS industrially -> which is one of the necessary preconditions (?) to gain high porosity of AFS that has(?) high metallurgical bonding strength. However, many detailed technical parameters of preparation of AFS still need to be determined before upscaling (native speaker needed) no any -> no OR not any missing: foaming background and potential future use of AFS , chemical reactions (if known), what materials are usually used, explain where science is with different rolling processes (cold rolling, hot rolling – there are many articles)
EXPERIMENTAL (English!)
common commercial metal element -> which metal element(s)? (manufacturer etc.) ° C -> °C (no space between ° and C) flat tubular cavity with a thickness of 21 mm: I presume this is the thickness of the wall of the tube? circular aluminum alloy tube φ 90 ´ 84 mm: is φ radius, if it is, then x between numbers is wrong and it should be 90-84 mm; if this is not radius and x means multiply, then unit is wrong, and it should be mm2) dilute hydrochloric acid -> how dilute (1 M, 1 g/L etc.) until the inner surface restored to light: what do you mean by that? Figure 1: text can be slightly bigger; powers->powders (missing letter d); it would be nice if all schemes in Figure 1 would be »designed« »identically«, i.e. same colour of the background also on the heated sketch (I understand why the plate is »reddish«, just not the background), there is no mention in article why one sketch is in rectangle (4th), one in dashed rectangle (1st) and the rest with no rectangles; Table 1: mesh is not explained in the text (if this is part of which metal elements were used, maybe mention that in the beginning of experimental »matrix powders for preparing aluminium foam were composed of common commercial metal element«, otherwise in Introduction among usual materials used?) both ports of flat tubular cavity -> both ends(?) of flat tubular cavity what kind of sponge was at the ends of the cavity preventing loss of sand? the powder was flowability for the rich of air inside: I do not understand that (did you mean that powder had still a lot of gaps so that air in the mixture could easily »flow« in it?) With more rolling passes and lower rolling rate (<5%): how much rolling passes and define rolling rate (is this rolling speed of the Al alloy tube divided by ?, since units are % - other articles define speed in mm/s (DOI: 1016/j.jma.2015.04.004)) Until extension along rolling direction occurs, the pre-rolling treatment stages accomplished: Did you mean: »When extension along rolling direction occurs, the pre-rolling treatment stage is accomplished?" the density of powder has reached the limitation manufactured: what is its number (of the density) thickness of production is about 16mm: is this thickness of the walls of tubes or height of the plate with the sand inside? why only those 3 temperatures were chosen: I understand why one would choose room temperature, but are 250 and 400 C somehow connected with properties of metals in powder form used? Are they connected with certain wanted crystal modifications or? If there is no special choice behind this 2 elevated temperatures, there is always good to find maximal value when final properties are still good (before the properties show drop), to find minimal temperature needed when final properties are still acceptable, or to find limit value, if the final properties would not decrease… - i.e. to find temperature dependence? (but ok…) cutting machine (EDM), the states above were observed by -> cutting machine (EDM), the samples were observed by (?) standard polishing techniques: which techniques? (http://eprints.nmlindia.org/1845/1/163-176.PDF) Combined with foaming test: foaming test is not explained (data can be found among results, but the foaming procedure should be explained also in the experimental part)
RESULTS and DISCUSSION (English!)
Figure 2c exhibited the sample that rolled at the temperature of 400 C. the -> sign for degree ° is missing, the should be The Figure 2. Section -> Figure 2. Cross-section? and the core layer was peeled by artificially -> and the core layer was peeled artificially?, how it was peeled (not mentioned in the experimental) in Figure 3b which experiencedmulti-pass rolling process: how many passes? Figure 4a show a set of experimental samples rolled at room temperature with 40% reduction per pass: what is reduced (the height of the tube filled with sand? – so is this “rolling reduction”, reduction of thickness, and how can it be bigger than 100 %? (max rolling reduction is defined as the radius of tube multiplied by the square of friction(?), did you mean by 40%, 80% and 120%, that reduction was 40 % at 1st pass, and you did 1, 2, 3 passes respectively? – because if you look at Figure 4 a or b, the thickness should be according to article 60 % of initial thickness, 20 % of the initial thickness (1/3 of the 60 % - which is not from the photos), and – 40 % (i.e. the material should be gone?) – i.e. rolling parameters should be more clearly defined in the experimental part: speed of rolling, radius of the tube, number of passes, and reduction of thickness can be measured and followed after every pass – the same goes for geometrical density, if this is the one observed (with errors included) – maybe even measure the pore distribution, size…) metallographic image of high power -> metallographic image of high magnification OR metallographic image made by means of a metallurgical microscope with high power? the plastic deformation of the powder particles seems more differently to achieve which is not conducive to filling of the gap between the particles -> sentence needs to be rewritten to be clear to the reader 54HV -> 54 HV (HV is unit; do the same for 27HV) Does in Figure 8 rolling rate become rolling elongation?, and is Expansion rate same as expansion factor E; is H0 initial height of precursor, i.e. before foaming, and H after foaming, i.e. what is aluminium foam layer and what is foamable precursor? Can you predict foaming dependence on numbers of rolling (do you see the limit value – i.e. would this change if you would roll over for 4, 5, 6… times more?) Can you predict foaming dependence on temperature of rolling (is there a limit value?; what is the highest temperature that can your “rolling machine” achieve?) However, the morphology was irregular, some of which also exhibited obvious crack-like bubble characteristics -> it looks like pores merged (you got open pores; with microCT you would be able to get more info through the whole volume; also you can use SEM and EDXS (mapping of larger area) to determine distribution of elements to help you conclude why once pores merge and once not – both before and after for further research (this is a lot of work which can not be done in 10 days); maybe you would also investigate foaming dependence on time on the temperature…) as shown in Figure 9c, a more uniform cell with better roundness morphology was obtained -> pores (bubbles) did not merge, you have closed pores (open pores can be used for acoustic insulation materials, closed pores for thermal insulation) Figure 9. Section diagram of foamed samples rolled at the rolling temperature -> of foamed samples rolled 3-times(?) at the rolling temperature higher density of powder core with excellent bonding state between powder particles -> higher density does not allow bubbles to escape because force keeping structure around the bubble together is stronger… For the precursor under similar foaming environment, more expansion rate and better cell morphology of the sample owned, the tightness combination condition of powder particles was -> this sentence is very hard to understand until the temperature rising up to the solid-liquid temperature -> was not mentioned in the experimental part
CONCLUSION (English!)
(2) It is hard to claim that “fluctuation of bonding interface« disappears completely when the temperature is above 400 C, when only 3 temperatures were tested (we know that fluctuations are present still at 250 C, but are gone at 400 C – for temperature in between we can not say anything)
(3) Densification process of core powder is not conducive to realize when the temperature was no more than 250℃ -> it is hard to understand the sentence
Density is mentioned many times, but without numbers. It should be defined which density is used in the article. Also, it should be stated, what is the lowest acceptable density and with which number of rollings according to the temperature used they reach the desired final density and other desired properties (if there are other), and what is the maximal density they reached and what is the optimal density reached (low energy input for good result – least needed number of rollings and lowest needed temperature). But if in the article authors want to show only that material can densify, than ok.

Author Response
Dear peer reviewer.
Thank you very much for your patience. After these days of hard work, numbers of deficiencies and errors in the article have been corrected under your kindly guidance. And also, the English expression was improved by one experienced English speaker.
The modifications are in the attached.

Round 2
Reviewer 2 Report
Dear authors. Your paper improved a lot, in the attachment are few minor recommendations of further improvement of the article, before you send it to English service for language corrections, although also English got a lot better. Kind regards

Author Response
Dear reviewers,
Thanks a lot for your valued comments. All the mistakes in this article that proposed has been corrected. Please see the attachment.
